# SEPT9 Upregulation in Satellite Glial Cells Associated with Diabetic Polyneuropathy in a Type 2 Diabetes-like Rat Model

**DOI:** 10.3390/ijms23169372

**Published:** 2022-08-19

**Authors:** Hung-Wei Kan, Yu-Cheng Ho, Ying-Shuang Chang, Yu-Lin Hsieh

**Affiliations:** 1School of Medicine for International Students, College of Medicine, I-Shou University, Kaohsiung 82445, Taiwan; 2School of Medicine, College of Medicine, I-Shou University, Kaohsiung 82445, Taiwan; 3Department of Anatomy, School of Medicine, College of Medicine, Kaohsiung Medical University, Kaohsiung 80708, Taiwan; 4School of Post-Baccalaureate Medicine, College of Medicine, Kaohsiung Medical University, Kaohsiung 80708, Taiwan; 5Department of Medical Research, Kaohsiung Medical University Hospital, Kaohsiung 80708, Taiwan

**Keywords:** diabetic polyneuropathy, septin-9, satellite glial cell, hypertriglyceridemia, neuropathic pain

## Abstract

Despite the worldwide prevalence and severe complications of type 2 diabetes mellitus (T2DM), the pathophysiological mechanisms underlying the development of diabetic polyneuropathy (DPN) are poorly understood. Beyond strict control of glucose levels, clinical trials for reversing DPN have largely failed. Therefore, understanding the pathophysiological and molecular mechanisms underlying DPN is crucial. Accordingly, this study explored biochemical and neuropathological deficits in a rat model of T2DM induced through high-fat diet (HFD) feeding along with two low-dose streptozotocin (STZ) injections; the deficits were explored through serum lipid, neurobehavioral, neurophysiology, neuropathology, and immunohistochemistry examinations. Our HFD/STZ protocol induced (1) mechanical hyperalgesia and depression-like behaviors, (2) loss of intraepidermal nerve fibers (IENFs) and reduced axonal diameters in sural nerves, and (3) decreased compound muscle action potential. In addition to hyperglycemia, which was correlated with the degree of mechanical hyperalgesia and loss of IENFs, we observed that hypertriglyceridemia was the most dominant deficit in the lipid profiles of the diabetic rats. In particular, SEPT9, the fourth component of the cytoskeleton, increased in the satellite glial cells (SGCs) of the dorsal root ganglia (DRG) in the T2DM-like rats. The number of SEPT9(+) SGCs/DRG was correlated with serum glucose levels and mechanical thresholds. Our findings indicate the putative molecular mechanism underlying DPN, which presumably involves the interaction of SGCs and DRG neurons; nevertheless, further functional research is warranted to clarify the role of SEPT9 in DPN.

## 1. Introduction

Diabetes mellitus is associated with high morbidity rates and substantial economic costs owing to its multiorgan complications, including neuropathy, retinopathy, and nephropathy. Of these complications, diabetic polyneuropathy (DPN) is the most common, affecting approximately 30% of patients with diabetes [1] who exhibit prominent and early neuropathic manifestations such as paresthesia and chronic pain [2,3]. Studies have focused on the molecular alterations of the dorsal root ganglia (DRG), particularly on intracellular signaling mechanisms underlying such alterations [4,5]. Regarding morphology, the degeneration of DRG nerve terminals (i.e., the intraepidermal nerve fibers [IENFs]) is a major pathological characteristic of DPN. IENF degeneration caused by nerve injury was proposed to be due to the disassembly of axoplasmic cytoskeletons [6]; this suggests that the role of cytoskeleton-related molecules in the development of neuropathic behaviors in DPN is worthy of investigation.

Septins, first discovered in yeast cells over 50 years ago, constitute the fourth component of the cytoskeleton. Septins are a highly conserved family of guanosine-5′-triphosphate-binding proteins that polymerize to form filaments in several vital biological processes [7,8,9]. Thirteen septin genes (*SEPT1* to *SEPT12* and *SEPT14*) have been identified in human tissues [10,11]. Studies have reported that septins exhibit high neuronal expression levels and play key roles in axonal transport, vesicular trafficking, neurotransmitter release, and neurological disease regulation; for example, SEPT9 is the causative gene in Charcot–Marie–Tooth disease [12], and its pathogenic mutations have been identified in hereditary neuralgic amyotrophy, an autosomal dominant neurological disorder [13,14,15]. However, the SEPT9 expression patterns in the DRG tissues and their roles in DPN have yet to be determined.

Satellite glial cells (SGCs) completely enclose the neuronal soma of the DRG. The literature indicates that SGCs play roles in neuroinflammation-mediated DPN by affecting nerve conduction and IENFs [16]. In addition, increased SGC numbers are correlated with chronic pain in streptozotocin (STZ)-induced type 1 diabetes mellitus (T1DM) [17], and SGC activation has been suggested to influence the excitation of DRG neurons [18], which contributes to neuropathic pain. Accordingly, investigating the SGC–DRG neuron interaction mechanism underlying DPN is essential.

Research on the molecular mechanisms underlying DPN has largely focused on T1DM animal models, namely rodent models of pancreatic β-cell damage induced by STZ, a cytotoxic glucose analogue [19]. However, DPN symptoms manifest earlier and are more prevalent in type 2 diabetes mellitus (T2DM) than in T1DM [20]. Several studies have employed T2DM-like diabetic models developed through a combined treatment of STZ with a high-fat diet (HFD), reporting elevated blood insulin levels and a further reduction of pancreatic β-cell function [21,22,23,24]; nevertheless, the mechanisms underlying DPN have not been fully explored, particularly the clinical implications of SEPT9′s role in T2DM.

To fill the aforementioned research gap, the present study developed a T2DM-like rodent model through a combined treatment of an HFD with two low doses of STZ. We then examined the alterations of serum biochemicals, neuropathic behaviors, IENF innervations of the skin, axonal morphometrics of the sural nerve, and motor nerve action potentials of the sciatic nerve in the T2DM-like rodent model. The major finding of the study is the upregulation of SEPT9 in SGCs under hyperglycemia, but the role of this in the development of DPN warrants further investigation.

## 2. Results

### 2.1. Physical and Biochemical Assessment in HFD/STZ and Control Group

The protocol of the HFD/STZ-induced T2DM-like rat model is listed in Figure 1A. To evaluate the effects of HFD and STZ coadministration, we measured the rats’ body weight and fasting blood glucose (FBG) levels weekly (Figure 1). We observed no difference in body weight between the control and HFD/STZ groups (Figure 1B). However, the HFD/STZ group exhibited a progressive and significant increase in FBG levels from week 5 (w5; 212.7 ± 30.4 vs. 121.0 ± 8.0 mg/dL, *p* < 0.01) after feeding protocols to week 10 (w10; 349.7 ± 28.3 vs. 109.2 ± 8.4 mg/dL, *p* < 0.001), the end of this study, compared with the control group, indicating fully developed diabetes (Figure 1C). Because the doses of STZ administered were low, no animal died in this study (Figure 1D).

We also examined serum lipid profiles (Figure 2), and the results revealed that the triglyceride (TG) level was significantly increased in the HFD/STZ group compared with the control group (*p* = 0.02), indicating hypertriglyceridemia after HFD/STZ administration. However, the total cholesterol (TC), low-density lipoprotein cholesterol (LDL-C), high-density lipoprotein cholesterol (HDL-C), and insulin levels in the HFD/STZ group were similar to those in the control group (Figure 2A). Because vascular risk factors have been identified as potential risk factors for neuropathy in individuals with diabetes [25], we also examined the TC:HDL-C ratio, a more critical risk factor than total or lipoprotein cholesterol for cardiovascular disease [26]. We observed that the TC:HDL-C ratio was increased in the HFD/STZ group compared with the control group (Figure 2B). Specifically, the TC:HDL-C ratio exceeded 5:1 in four out of nine rats (44.4%) in the HFD/STZ group.

### 2.2. Development of Mechanical Hyperalgesia and Reduced Free Navigation Activity in the HFD/STZ Group

To investigate neurobehavioral disorders, we performed the von Frey monofilament test for mechanical response and an open-field test involving a free navigation task. The rats in the HFD/STZ group became hypersensitive to tactile stimuli at w10, and this was evidenced by reduced mechanical thresholds compared with those in the control group (9.0 ± 4.5 vs. 41.9 ± 26.5 g, *p* = 0.01; Figure 3A). Furthermore, mechanical thresholds were inversely correlated with serum glucose levels (Figure 3B). In the open-field test, the rats in the HFD/STZ group exhibited fewer exploratory behaviors in the central box area (Figure 3C, D). Quantitative results revealed that the retention time in the central box area was significantly reduced in the HFD/STZ group compared with that in the control group (8.8 ± 10.7 vs. 22.8 ± 12.9 s, *p* = 0.03; Figure 3E). These data thus suggest that early features of DPN were associated with neuropathic manifestations and depression-like behavior; this may have resulted from neuropathological and neuroelectrophysiological disorders in the HFD/STZ group.

### 2.3. Neuropathological and Neuroelectrophysiological Characteristics in the HFD/STZ Group

We also examined neuropathological changes in the rats by assessing the IENF density in the skin and conducting a morphometric analysis of the sural nerves. We observed that in the control group, PGP9.5(+) IENFs populated the epidermis and dermis of the footpad skin, forming nerve bundles in the footpad skin. By contrast, in the HFD/STZ group, PGP9.5(+) IENFs were reduced in the epidermis of the skin with swollen and fragmented dermal fibers (Figure 4A). Our quantitative results revealed that the IENF density in the HFD/STZ group was significantly reduced compared with that in the control group (8.4 ± 1.7 vs. 12.8 ± 2.4 fibers/mm, *p* = 0.001; Figure 4B). However, these changes in IENF density were independent of the TG (Figure 4C) and serum glucose (Figure 4D) levels. The results of our morphometric analysis of the sural nerves (Figure 5A, B) revealed that the control and HFD/STZ groups exhibited similar myelinated fiber densities (1851.0 ± 315.3 vs. 1914.0 ± 158.4 axons/mm^2^, *p* = 0.73; Figure 5C); however, the HFD/STZ group exhibited a decrease in axonal size (25th–75th percentile: 3.3–14.9 vs. 4.1–16.9 µm^2^, *p* < 0.001; Figure 5D) and mean axonal diameter (3.4 ± 0.1 vs. 3.6 ± 0.1 µm, *p* = 0.007; Figure 5E). These findings suggest that both unmyelinated and small-to-medium myelinated sensory axons are affected early during the course of neuropathy. Regarding neuroelectrophysiological characteristics, we noted that the CMAP amplitudes were significantly reduced in the HFD/STZ group compared with the control group (12.3 ± 9.8 vs. 26.8 ± 3.9 mV, *p* = 0.02; Figure 6). Overall, these findings resemble the clinical manifestations of DPN.

### 2.4. Increased SEPT9 in SGCs in the HFD/STZ Group

Notably, this study demonstrated a novel SEPT9 expression pattern in the DRG of the rats in the HFD/STZ group (Figure 7). We observed that SEPT9(+) cells, especially SGCs, increased in the HFD/STZ group (Figure 7A). Our quantitative results indicated that the number of SEPT9(+) SGCs per DRG were increased in the HFD/STZ group compared with those in the control group (7.4 ± 0.6 vs. 5.5 ± 1.7 SGCs/DRG, *p* = 0.03; Figure 7B). In particular, DRG neurons preferentially enclosed by the SEPT9(+) SGCs were observed to have a smaller diameter in the HFD/STZ group than in the control group (39.2 ± 4.6 vs. 45.5 ± 3.0 μm, *p* = 0.02; Figure 7C). For example, the histogram spectrum of DRG neuronal sizes with SEPT9(+) SGCs exhibited a left shift (Figure 7D). Although the numbers of SEPT9(+) SGCs/DRG were not correlated with IENFs (Figure 7E), they exhibited a linear correlation with the serum glucose levels (*r* = 0.72, *p* = 0.008; Figure 7F) and mechanical thresholds (*r* = −0.70, *p* = 0.02; Figure 7G).

## 3. Discussion

This study developed a T2DM-like rat model through an HFD/STZ coadministration protocol. The model exhibited DPN features consistent with consensus criteria [27], including decreased IENF densities, reduced myelinated axonal diameters, declined CMAP amplitudes, and marked mechanical hyperalgesia. We also demonstrated depression-like behavior in the HFD/STZ group, which is in agreement with the finding of a previous study that suggested an association between persistent pain and mood disorders in patients with diabetes [28]. Furthermore, we revealed that SEPT9, a novel molecule, was upregulated in DRG SGCs and correlated with hyperglycemia and mechanical hyperalgesia.

Although we sought to induce obesity by using an HFD, our STZ injection appeared to exert a greater effect on body weight than did the HFD; that is, discrepancies in body weight were observed for different STZ administration strategies [29]. For example, the HFD/STZ group did not exhibit dramatic changes in body weight. This manifestation may be caused by the natural cytotoxicity of STZ, which results in rapid β-cell failure or increased leptin and adiponectin levels in the diabetic rats [30], mimicking the natural pathophysiology of lean patients with diabetes, who typically exhibit more severe functional insulin secretory defects than do obese individuals [31]. Alternatively, because the fat percentage to total body weight is more important than total body weight, excess adiposity with decreased muscle mass should be assessed [31].

We also examined lipid profiles and found that our HFD/STZ protocol induced only significant hypertriglyceridemia without other forms of dyslipidemia. Although hyperlipidemia has been demonstrated in other HFD/STZ protocols [23], serum TC and LDL-C levels seem to be affected by the duration of high-fat diets and dosages of STZ [25,32]. In contrast, hypertriglyceridemia consists constant defects among various HFD/STZ protocols [24]. Serum triglyceride levels constitute a crucial neuropathic factor and have been reported to be associated with painful DPN [20], lower extremity amputation [33], and neuropathy progression [34]. In addition, studies on T2DM have demonstrated triglycerides to be a risk factor for early diabetic neuropathy [35] and to be closely correlated with IENF densities [35]. Overall, our lipid profile analysis results reveal the role of marked hypertriglyceridemia in the early stages of HFD/STZ-induced diabetes in our animal model; hence, we postulate that hypertriglyceridemia may influence the development of HFD/STZ-induced DPN.

Regarding the molecular alteration in the DRG of the HFD/STZ group, we observed that SEPT9 was upregulated in the SGCs, especially in enclosed neurons with smaller diameters. Gene alterations in SGCs after nerve injury have been implicated in the promotion of axon regeneration [36], which also predisposes DRG neurons to nociceptive feedback by increasing neuronal excitability through glutamate expression [37]. SEPT9 has been shown to be an upstream regulator of the N-methyl-D-aspartate receptor subunit NR2B on SGCs [38], which may be involved in the generation of mechanical nociception [39]. In addition to being upregulated in SGCs, SEPT9 is upregulated by the low-stiff extracellular matrix [40] owing to the imbalance between matrix formation and matrix degradation in diabetes [41]. Therefore, investigating the extent and nature of the microenvironments surrounding DRG neurons and SGCs, which may be affected by hyperglycemia-induced chronic inflammation and tissue fibrosis, is warranted. Despite the blood-nerve barrier being weak in the DRG, targeting SGCs is still an easier approach to reduce neuronal hypersensitivities during diabetic neuropathic pain based on the nature of the SGCs which communicate with the perineuronal environment. On the basis of the literature and our findings, we propose that this mechanism may in part cause the progression of DPN. Further functional characterization is necessary to determine the possible therapeutic role of SEPT9.

## 4. Materials and Methods

### 4.1. Generation of T2DM-like Diabetic Rodent Model through HFD and STZ Induction

Male Sprague–Dawley rats (age, 6 weeks; weight, 176–200 g; LASCO, Taipei, Taiwan) were used, and they were adapted for at least 5 days (12/12-h light/dark cycle, 22–24 ℃, 40–60% humidity). The animals were then assigned randomly to two dietary treatment groups: a control group (*n* = 9) and an HFD/STZ group. The control group comprised rats that received a standard chow diet (SCD; LabDiet, St. Louis, MO, USA) for 3 weeks, followed by 10 mM sodium citrate buffer (0.25 mL/kg, pH 4.5) administered intraperitoneally. The HFD/STZ group comprised T2DM-like diabetic rats induced through the administration of an HFD (D12492, Research Diets, New Brunswick, NJ, USA) for 3 weeks, followed by the intraperitoneal administration of STZ (Sigma-Aldrich, St. Louis, MO, USA) at 25 mg/kg weekly for 2 weeks (Figure 1A). The SCD or HFD was administered from the beginning (week 0, w0) through week 10 (w10) of the feeding protocol throughout the study period. FBG levels were measured weekly by using a blood glucose meter (GS550, Bionime, Taichung, Taiwan). Rats with FBG levels of >250 mg/dL were considered to be diabetic. All study procedures were approved and conducted in accordance with the ethical guidelines for laboratory animals established by the Institute Animal Care and Use Committee at I-Shou University (#ISU-109007) and Kaohsiung Medical University (#110115), Taiwan.

### 4.2. Neurobehavioral Assessment

Von Frey monofilament test: The up-down method was applied to determine mechanical thresholds by using the von Frey monofilament test [42]. The rats were individually placed in plastic cages with a metal mesh and allowed to habituate for 30 min. The monofilament was applied perpendicularly to the plantar surface of the hind paw until it buckled, delivering a constant predetermined force for 2–5 s. A response was considered to be positive if the rat exhibited either brisk withdrawal or paw flinching during the monofilament application. A 100 g monofilament was selected as a cutoff point to prevent tissue injury. The mechanical thresholds were then calculated using the established formula [42].

Free navigation in open-field arena: The rats were individually placed in the center of a black arena (120 × 120 × 120 cm). The times spent by the tested rats in the defined central zone and in the remaining zone were recorded for 10 min, and their locomotion activities were analyzed using EthoVision XT tracking software (Noldus Information Technology, Wageningen, The Netherlands).

### 4.3. Serum Analysis

Whole blood was collected and then centrifuged for 10 min (2500× *g*, 4 °C). Serum glucose, TG, TC, LDL-C, and HDL-C levels were measured (National Laboratory Animal Center, Taipei, Taiwan). Moreover, serum insulin levels were assessed through an enzyme-linked immunosorbent assay (ELISA) in accordance with the manufacturer’s manual (Rat Insulin Kits; Elabscience, Houston, TX, USA). The insulin concentration was measured at 450 nm on an ELISA reader (ELx-800, BioTek Instruments, Winooski, VT, USA) and derived using the corresponding standard curve.

### 4.4. Nerve Conduction Study

Compound muscle action potential (CMAP) was recorded using Biopac Systems (Goleta, CA, USA) in accordance with our previously described protocols [43]. Monopolar stimulating electrodes were inserted at the sciatic notch, and recording electrodes were placed on the plantar muscles. Stimuli were applied at an intensity of 100 V and a duration of 0.1 ms. The mean amplitudes were measured from the baseline to the peak of each CMAP waveform and then analyzed.

### 4.5. Immunohistochemical Analysis and IENF Innervation Quantification

Footpad skin tissues were fixed through intracardiac perfusion with 2% paraformaldehyde fixative and stored in 0.1 M phosphate buffer (PB, pH 7.4). Immunohistostaining was conducted, and the IENF innervations were quantified in accordance with established protocols [43]. Briefly, the footpad skin was sectioned in 30-μm thickness perpendicularly to the plantar surface and incubated with anti-protein gene product 9.5 (PGP9.5; rabbit, 1:500; Abcam, Cambridge, MA, USA) antiserum, a panaxonal marker. PGP 9.5(+) IENFs in the epidermis were counted, and the length of the epidermis along the lower margin of the stratum corneum in each section was measured using Image-Pro Plus software (Media Cybernetics, Rockville, MD, USA). The IENF density was calculated (in fibers/mm).

### 4.6. DRG Immunofluorescence Staining

The molecular expression profiles on the fourth- and fifth-lumbar DRGs were assessed using a regular immunofluorescence approach, as described in our previous study [5]. In the present study, the primary antisera were anti-SEPT9 (rabbit, 1:100; Proteintech, Rosemont, IL, USA) and anti-non-phosphorylated neurofilament (mouse, 1:1000; Biolegend, San Diego, CA, USA). Fluorescein-isothiocyanate- and Texas-Red-conjugated secondary antisera were applied for the dual-labeling immunostaining of SEPT9 and the nonphosphorylated neurofilament, respectively.

### 4.7. Sural Nerve Morphometry Quantification

Sural nerves were fixed with 5% glutaraldehyde in 0.1 M PB for 2 h and then embedded with Epon 812 resin (Polyscience, Philadelphia, PA, USA). Sural nerve pathology was assessed in accordance with our previously established protocol [43]. Semithin sections (0.5 μm) were cut with an ultramicrotome (Reichert Ultracut E, Leica, Wetzlar, Germany) and stained with toluidine blue. For morphometric analyses, the sections were photographed at 400× magnification under an ECLIPSE Ci-S microscope (Nikon, Tokyo, Japan). Image-Pro Plus software (Media Cybernetics) was used to calculate axonal density (axons/mm^2^), mean axonal diameter (μm), and axonal area distribution, as previously described [43].

### 4.8. Data Analysis

Data were analyzed with Graphpad Prism ver. 9.4.1 (Graphpad Software, San Diego, CA, USA) and are presented herein as mean ± standard deviation. For comparisons between controls and diabetic values at each time point, two-tailed unpaired *t* tests or multiple *t* tests with the Holm–Sidak correction were performed. For linear regression analysis between variables, Pearson’s correlation with the slope of the regression line, including the 95% confidence interval, was used. *p* < 0.05 was considered statistically significant.

## Figures and Tables

**Figure 1 ijms-23-09372-f001:**
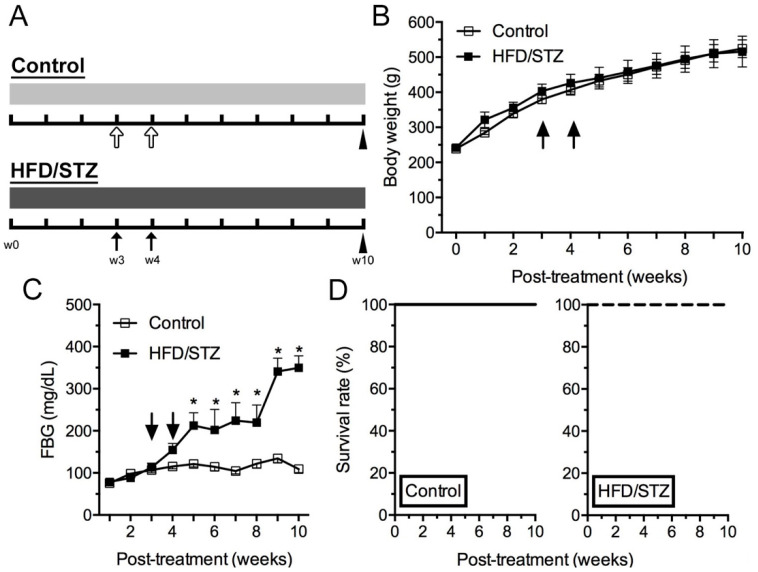
Experimental design and assessments of body weight and fasting blood glucose (FBG). (**A**) Rats were assigned to the control (upper panel) and high-fat diet (HFD)/streptozotocin (STZ) groups (lower panel). Light gray bar indicates rats fed a standard chow diet and dark gray bar indicates those fed an HFD. Either citrate buffer (hollow arrow in upper panel) or STZ (solid arrow in lower panel) were administered at week 3 (w3) and week 4 (w4) after the beginning of feeding protocol (w0). Arrowhead indicates end of experiments (w10). (**B**,**C**) Graphs indicating the changes in body weight (**B**) and FBG (**C**) between control (opened square) and HFD/STZ (filled square) groups. Arrows: STZ injections. * *p* < 0.05: compared with control group. (**D**) Comparison of survival curves between control (left panel) and HFD/STZ groups (right panel). No experimental animals died in this study.

**Figure 2 ijms-23-09372-f002:**
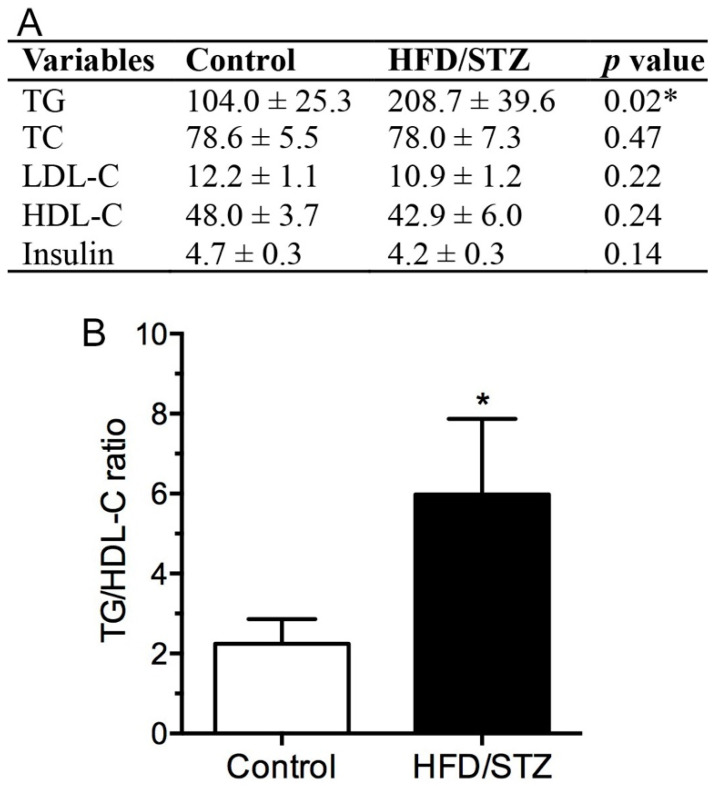
Serum assay results for the HFD/STZ group. (**A**) Biochemical assay conducted using serum from the experimental rats. HFD, high-fat diet; STZ, streptozotocin; TG, triglyceride; TC, total cholesterol; LDL-C, low-density lipoprotein cholesterol; HDL-C, high-density lipoprotein cholesterol. Unit: mg/dL for TG, TC, LDL-C, and HDL-C; ng/dL for insulin. * *p* < 0.05. (**B**) Graph indicating differences in TC:HDL-C ratio. * *p* < 0.05.

**Figure 3 ijms-23-09372-f003:**
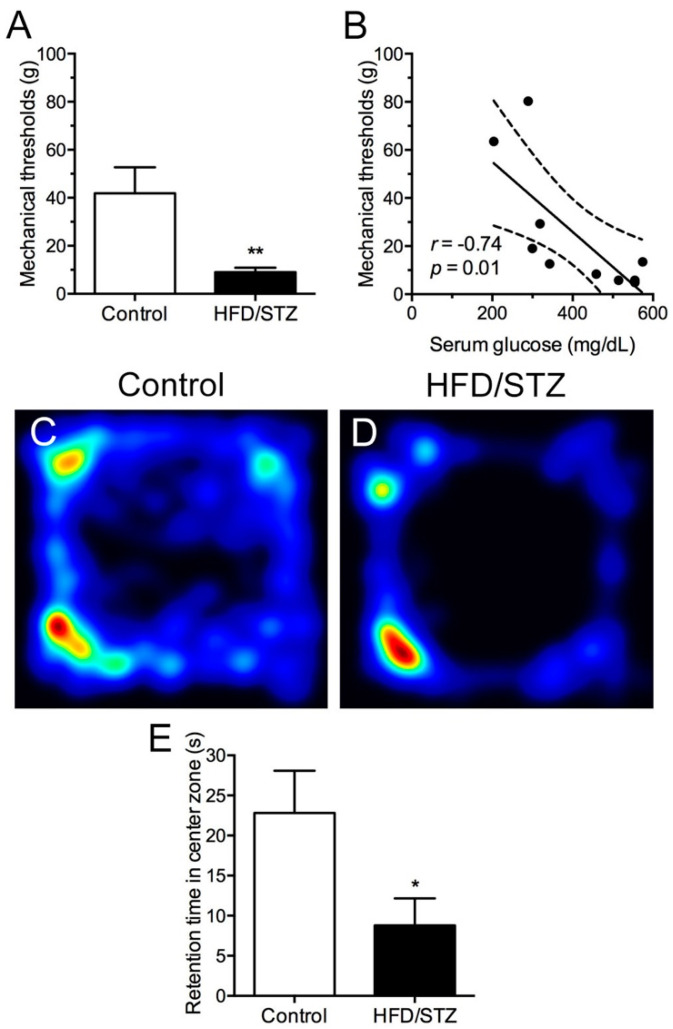
Neurobehavioral changes in the HFD/STZ group. Neurobehavioral assessment conducted using the von Frey monofilament test for mechanical response and an open-field test involving free navigation, as described in the Materials and Methods section. (**A**) The mechanical threshold of the HFD/STZ group was significantly reduced, and (**B**) the mechanical thresholds were inversely correlated with serum glucose levels. ** *p* < 0.01. (**C**,**D**) Graphs illustrating the navigation heating trace in an open field for the (**C**) control and (**D**) HFD/STZ groups. (**E**) Bar graph indicating differences in navigation time in the central area of the open field. * *p* < 0.05. HFD, high-fat diet; STZ, streptozotocin.

**Figure 4 ijms-23-09372-f004:**
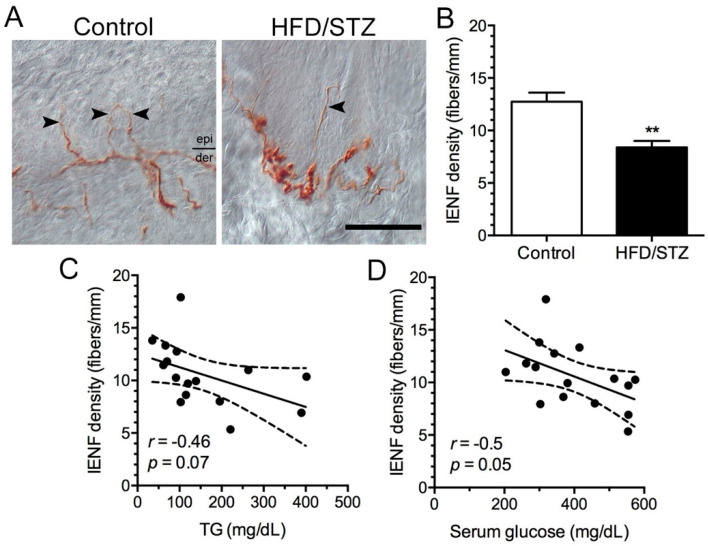
Changes in intraepidermal nerve fibers (IENFs) in the HFD/STZ group. (**A**) IENFs of the skin were observed using protein gene product 9.5 (PGP9.5)-immunostaining in the control (left panel) and HDF/STZ (right panel) groups. PGP9.5(+) IENFs (arrowhead) were decreased in the HFD/STZ group. Bar, 50 μm. (**B**) Bar graph illustrating the quantitated PGP9.5(+) IENFs. ** *p* < 0.01. (**C**,**D**) Graphs indicating the linear correlations of PGP9.5(+) IENFs with (**C**) TG and (**D**) serum glucose. HFD, high-fat diet; STZ, streptozotocin; TG, triglyceride.

**Figure 5 ijms-23-09372-f005:**
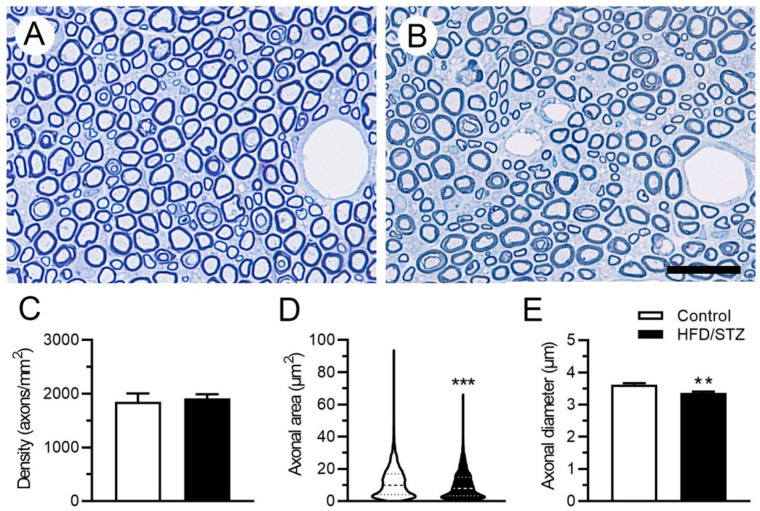
Morphometrics of sural nerves in the HFD/STZ group. (**A**,**B**) Photographs depicting semithin sections of Epon-embedded sural nerves in the control (**A**) and HFD/STZ groups (**B**). HFD, high-fat diet; STZ, streptozotocin; Bar, 25 μm. (**C**,**D**) Morphometric analysis results for sural nerves, revealing changes in axonal density (**C**) and axonal area (**D**). The dashed line represents the median percentile, and the dotted line represents the 25th and 75th percentiles. *** *p* < 0.001. (**E**) Graph illustrating differences in mean axonal diameter of sural nerves. ** *p* < 0.01.

**Figure 6 ijms-23-09372-f006:**
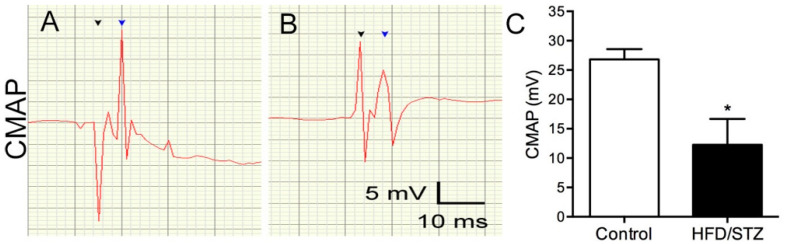
Reduced amplitudes of motor nerve action potential in the HFD/STZ group. (**A**,**B**) Graphs illustrating the waveforms of the compound muscle action potential (CMAP) at week 10 for the control (**A**) and HFD/STZ (**B**) groups. HFD, high-fat diet; STZ, streptozotocin; black arrowhead, stimulation; blue arrowhead, response. (**C**) Graph showing the difference in CMAP amplitudes between the two groups. * *p* < 0.05.

**Figure 7 ijms-23-09372-f007:**
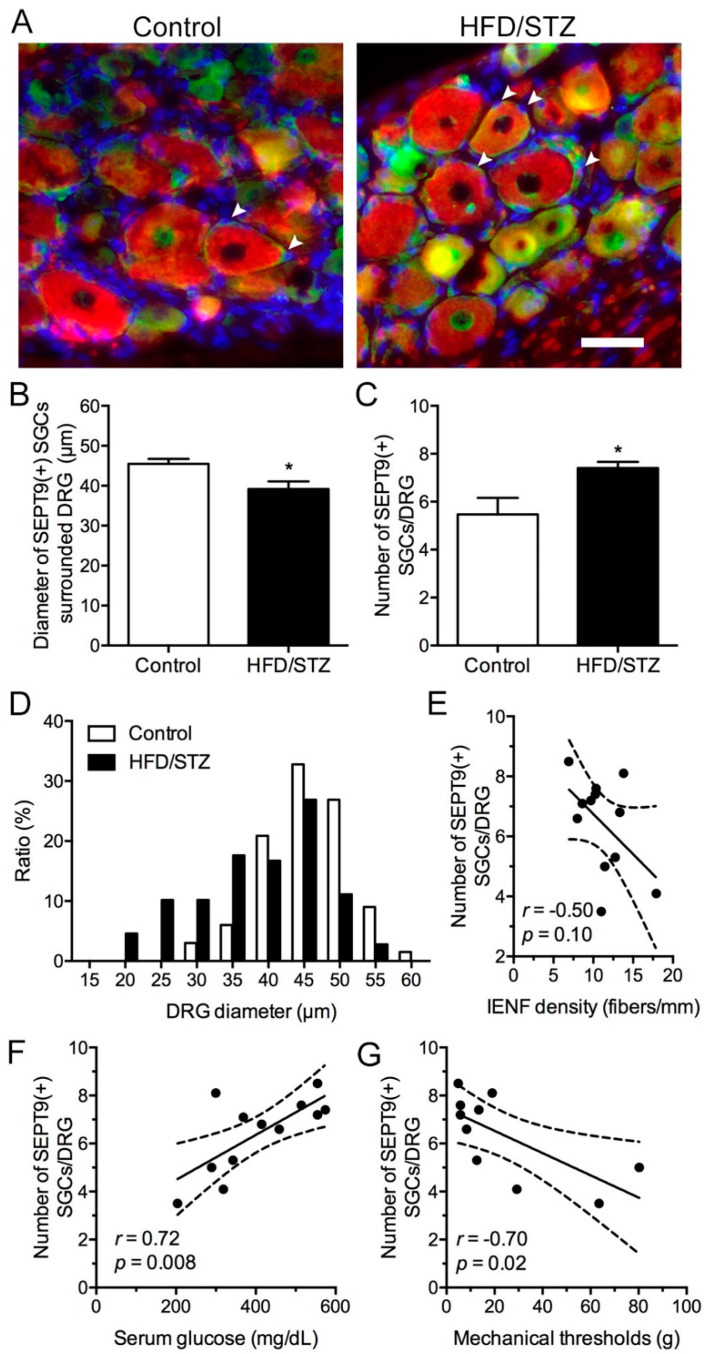
Neuropathology of SEPT9(+) satellite glial cells (SGCs) in the HFD/STZ group. (**A**) Double-labeling immunostaining of SEPT9 (in green) and nonphosphorylated neurofilaments (in red) in the dorsal root ganglion (DRG) of the control (left panel) and HDF/STZ groups (right panel). HFD, high-fat diet; STZ, streptozotocin. Arrowhead indicates SEPT9(+) satellite cells. Bar, 50 μm. (**B**–**D**) Graphs showing the quantified numbers of SEPT9(+) SGCs/DRG (**B**), and morphometric analysis results for the neuronal size (**C**) and histogram spectra of SEPT9(+) SGCs/DRG neurons (**D**) in the control (hollow bar) and HFD/STZ (solid bar) group. * *p* < 0.05. (**E**–**G**) Graphs showing the correlation of SEPT9(+) SGCs/DRG with IENF density (**E**), serum glucose (**F**), and mechanical thresholds (**G**). IENF, intraepidermal nerve fiber.

## Data Availability

The data that support the findings of this study are available from the corresponding author upon reasonable request.

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
