# Peer review of "SEPT9 Upregulation in Satellite Glial Cells Associated with Diabetic Polyneuropathy in a Type 2 Diabetes-like Rat Model"

_ijms, 2022, doi:10.3390/ijms23169372_

Round 1

Reviewer 1 Report

This is a well-written manuscript discussing the title “SEPT9 upregulation in satellite glial cells associated diabetic polyneuropathy” The manuscript certainly contributes new information, in terms of characterizing some pathological mechanisms involved in disease development (diabetic polyneuropathy). I just have a few comments, to be further considered to enhance quality of the manuscript. 

Specific comments

Within the title, I recommend authors add/implement a model (rats) being used for the project. This will make the presentation more accurate

Please add citations for the statement “Research on the molecular mechanisms underlying DPN has largely focused on T1DM animal models…”

Why did you not see any difference with the cholesterol levels?

State which software was used for statistical analysis? Perhaps also indicate why only used t-test as this may be an add on test, supplementing other accepted comparisons like Tukey and so on…

Author Response

Reviewer 1 comments:

This is a well-written manuscript discussing the title “SEPT9 upregulation in satellite glial cells associated diabetic polyneuropathy” The manuscript certainly contributes new information, in terms of characterizing some pathological mechanisms involved in disease development (diabetic polyneuropathy). I just have a few comments, to be further considered to enhance quality of the manuscript. 

Response: We appreciate your valuable comments and suggestions and have made revisions accordingly.

Specific comments

Within the title, I recommend authors add/implement a model (rats) being used for the project. This will make the presentation more accurate

Response: Thanks for the valuable suggestion. We modified the title accordingly.

P1, Line3, “Title”:

SEPT9 upregulation in satellite glial cells associated diabetic polyneuropathy in a type 2 diabetes-like rat model

Please add citations for the statement “Research on the molecular mechanisms underlying DPN has largely focused on T1DM animal models…”

Response: Thanks for the comments. We have added the reference in the revised text.

P12, line 402-3, “Reference” section:

  1. Pandey, S.; Dvorakova, M. C., Future Perspective of Diabetic Animal Models. Endocr Metab Immune Disord Drug Targets 2020, 20, 25-38.

Why did you not see any difference with the cholesterol levels?

Response: Thanks for the critical comments. We have added explanation in the Discussion.

P9, line 229-32, “Discussion” section:

Although hyperlipidemia have been demonstrated in other HFD/STZ protocols [23], serum TC and LDL-C levels seem to be affected by the duration of high-fat diets and dosages of STZ [25, 32]. In contrast, hypertriglyceridemia consists constant defects among various HFD/STZ protocols [24].

State which software was used for statistical analysis? Perhaps also indicate why only used t-test as this may be an add on test, supplementing other accepted comparisons like Tukey and so on…

Response: Thanks for the valuable suggestions. We added the statistic details accordingly.

P11, line 338-43, “Materials and Methods” section:

Data were analyzed with Graphpad Prism ver. 9.4.1 (Graphpad Software, LLC.) and are presented herein as mean ± standard deviation. For comparisons between controls and diabetic values at each time point, two-tailed unpaired t-tests or multiple t-tests with the Holm-Sidak correction were performed. For linear regression analysis between variables, Pearson’s correlation with the slope of the regression line, including the 95% confidence interval, was used. p < 0.05 was considered statistically significant.

Reviewer 2 Report

This is an interesting manuscript that certainly adds to the present knowledge about diabetic neuropathy. I would only suggest expanding Discussion a bit, including the hypothetical clinical consequences that these findings may have as well as potential therapeutic approaches that might have positive influence on reversing of the changes.  

Author Response

Reviewer 2 comments:

This is an interesting manuscript that certainly adds to the present knowledge about diabetic neuropathy. I would only suggest expanding Discussion a bit, including the hypothetical clinical consequences that these findings may have as well as potential therapeutic approaches that might have positive influence on reversing of the changes.

Response: We very much appreciate your suggestion and have incorporated changes in the revised Discussion.

P9, line 252-55, “Discussion” section:

Despite blood-nerve barrier is weak in the DRG, targeting SGCs is still an easier approach to reduce neuronal hypersensitivities during diabetic neuropathic pain based on the nature of SGCs which communicate with the perineuronal environment.
